# The Spiral of Escalating Water Conflict: The Theory of Hydro-Politics

**Sameh W. H. Al-Muqdadi**

Green Charter GC., Franz-Belzer Str. 2, 76316 Malsch, Germany; sameh@green-charter.de

**Abstract:** Using water to enforce a political agenda is a global concern for peacebuilding. Hence, understanding hydro-politics is essential when predicting possible water-based conflict scenarios between riparian countries. A structured theory covering most of the possible events involved in hydro-politics would help assess with a sufficient understanding the reasons and consequences of water conflict. This study proposed a comprehensive theory of hydro-politics, particularly those related to water impoundment and water control through upstream country dams, to identify the root causes of water conflicts between riparian states and the factors of global challenges that arise in conflicts. The framework used eight phases elaborated on the key theories of international relations and demonstrated the possible connection between water conflict/cooperation events and the adopted international relations doctrine at the state level. Each phase illustrates the hydro-political relations between the riparian countries, expected level of conflict, power balance, and possible consequences. Additionally, 21 international case studies were used to illustrate these phases. The theory may assist decision makers in analyzing collective risk and alleviating any expected negative implications of water conflicts.

**Keywords:** water conflict; hydro-politics; international relations; water policies; water governance

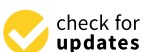



## 1. Introduction

Scientific efforts have been made over the last couple of decades to theorize the hydro-political aspects and the possibility of water conflicts. Different tools have been used to serve that objective, and the literature has intensively examined transboundary watercourses. The international relations theory framework is one of the tools employed to understand and criticize crucial elements that may shape the behavior of the state. Such an understanding would help to disclose the root cause of conflict and cooperation over transboundary water. However, a comprehensive and collective theory of hydro-politics based on the dam impact and controlling water by the upstream country is still missing, although having such a structured theory would help decision makers to realize the cost of non-cooperation versus the privileges of strategic agreements over transboundary water.

### 1.1. Theorizing the Hydro-Politics

Early efforts to discuss hydro-politics date back two decades to Turton et al., 2003 [1], who argued that there was no adequate hydro-political theory and proposed three hypotheses focusing on the Southern African hydro-political complex and an expanded definition of hydro-politics. This provided insights into the evaluation of case studies and helped others develop hypotheses for Southern Africa and other regions. Turton, in 2006 [2], discussed the concept of hydro-hegemony and hydro-politics complex theory based on international cases. This study included key questions related to water hegemony, with the goal of developing hydro-politics as a discipline by considering the expected drivers. These results define the hydro-political complex for Southern Africa by sorting the riparian countries into two groups, pivotal and impacted states, which represents an early initiative to establish a hydro-politics theory that defines water system complexity and dynamics

via securitization and politicization. This study recommended further development of the discipline to understand the complex hydro-political system. Conca, in 2006 [3], discussed the performance of water governance at the local level and its impact on the water war looming and claimed that ignoring the acute need to govern and manage water resources locally would escalate it to the regional and global levels. The author believes that disputes over water resources will be stalled due to the multiple functions of water. Later, the author [4] encouraged the concept of regional water governance approaches instead of global environmental governance. The author considered four sources of motivation for his hypothesis: the regional approach has the privileges of offering hope in terms of political progress, less complication in resource management, being more propitious to promoting new norms, and considering an essential foundation for building global environmental governance. Cascão and Zeitoun, in 2013 [5], introduced a theory consisting of four stages that reflected the key factors in hydro-hegemony used to control transboundary flows: geography, resource levels, bargaining power, and ideational power. This study considered transboundary waters spanning political economies on national, regional, and international scales. In particular, the Middle East–North Africa (MENA) region was considered a case study. Their theory posited valuable outcomes if non-hegemonic actors were involved, whereas mild hegemony led to bargaining and resource power being more important than geographical position or ideational power. Kehl, in 2011 [6], examined riparian countries' asymmetrical power and evaluated the adoption of cooperative versus conflict strategies. This research was grounded in a theoretical framework for hydro-hegemony associated with hard and soft power that considered the influence of geographic, economic, political, military, technological, and external factors on water governance. Weak players were reported to invest in the development of their capacity to cope with hydro-political systems. Soft power and direct dialogue were found to have a better chance of success, whereas the use of hard power was ineffective. The author also emphasized that multinational corporations and actors could maintain leverage in hydro-political complexes. Madani et al., in 2011 [7], demonstrated a non-cooperative game theory concept using the Nile River as a case study to investigate the options for cooperation among riparian countries in terms of water allocation. The numerical graph model for conflict resolution (GMCR II) decision support system was implemented, the model boundaries were determined, and riparian countries were given three options: stability, sensitivity to downstream or upstream countries, and instability. The results identified six equilibrium states and high sensitivity to the preferences of downstream countries, and conflict was predicted to be solved peacefully with political and economic pressure. Mirumachi, in 2015 [8], emphasized that the reviewing of just conflict or cooperation offers a limited picture of international transboundary waters and riparian relationships. The author believes in the lacked robust theorization in dealing with transboundary waters, where the description of the riparian interaction as a process that involves a range of actors is essential to examine the complex political system of controlling, negotiating, and governing shared water resources. The research presented The Transboundary Waters Interaction NexuS (TWINS) as a comprehensive conceptual framework that analyzes conflict and cooperation simultaneously through the evolution of relationships between basin states. Swyngedouw, in 2015 [9], proposed a series of theoretical tools to debate political ecology. Despite the variety of epistemological perspectives on international relations theories, the author emphasized that there is an urgent need to depoliticize environmental matters and explore the political nature of the ecological process. However, Swyngedouw, in 2017 [10], concluded that the political dimension was almost impossible to foreclose. Shahbazbegian et al., in 2016 [11], adopted theoretical hydro-political drivers and visualized them via system dynamics to illustrate the time impact. Their study mapped hydro-political self-organization (HSO) using the Helmand River as a case study, with five groups of hydro-political variables representing watershed and water transboundaries: primary, third-party, national, regional, and state-building. The authors analyzed the gaps in technical studies and combined this with an empirical process, assuming that the HSO can be integrated with system policies as a useful tool

to model the hydro-politics of a basin over time. Rai, Wolf, and Sharma, in 2017 [12], compiled a hydro-diplomacy database of events and actions related to water cooperation and conflict from 1874 to 2014 from a wide range of sources focusing on the Ganges basin between India and Nepal. This study evaluated the intensity of these incidents on spatial and temporal scales. The bi-national water event method was used to explore water events, with cooperative events being much more prevalent (92%) than conflict or neutral events (8%). Thus, the area was subject to moderately positive cooperation; however, the authors argued that riparian countries needed to reconsider their water transboundary relations and develop joint regional socioeconomic projects, critical from a geopolitical perspective for both countries. Zeitoun et al., in 2017 [13], proposed a conceptual framework to examine the dynamic behavior of the state in terms of water transboundary hegemony. This study used integrated theories of change and counter-hegemony to determine the mechanisms associated with water transformation. The results showed that most hydro-political interactions involved both compliance and competition. Furthermore, power asymmetry influenced hydro-political strategies and fed the behavior of the state. The authors presented a summary of non-hegemonic basins and transboundaries on a global scale. The framework also illustrates how transboundary water events were associated with sociopolitical processes. Farnum, in 2018 [14], investigated a traditional fog-harvesting system in local areas of Morocco and how it was integrated with scientific innovation. The author argued that the concept of water diplomacy as a theory was still affected by the biases of its source disciplines and that a complex water conflict/cooperation system required holistic expertise and policy interventions. This research employed track diplomacy theory to illustrate how fog water was used for peacebuilding and social solidarity. The research emphasized that theorists and decision makers must review and evaluate previous mechanisms and search for initiatives in water diplomacy as alternatives to conventional approaches by global water peacemakers. Grech-Madin et al., in 2018 [15], offered tools for water diplomacy at the inter-state level and investigated political norms. The tool is assumed to be valuable for developing an understanding of water practices and promoting effective policies using the African continent as a case study to map the risk of water conflict. Islam and Susskind, in 2018 [16], used an analytical approach based on negotiation theories to model the complexity of water challenges and conflicts. The author introduced three negotiation theories: fact-finding, value creation through option generation, and stakeholder identification and engagement. The research described the factors affecting water conflicts as a complex system crossing multiple boundaries and emphasized that any resolution would require interactions between multiple sectors. The authors argued that modeling was a difficult task, but negotiation theories made it easier to address the supply–demand gaps in complex water problems. It was also posited that the various dimensions associated with water conflicts should be considered together and could not be separated. Petersen-Perlman and Fischhendler, in 2018 [17], examined and conceptualized power dynamics in transboundary hydro-politics, with power often misused within a hegemonic system and structure. This study assumed that a non-hegemonic party was often unable to achieve positive outcomes. The authors argued that understanding hegemonic vulnerabilities was more productive than interpreting the dynamics of transboundary water. Identifying the weaknesses of powerful actors gives the non-hegemonic party leverage to achieve better power distribution. Farinosi et al., in 2018 [18], presented a predictable conceptual model to estimate hydro-political interaction trends. The authors highlighted common factors leading to water conflicts across political boundaries, particularly changes in socioeconomic and biophysical conditions. The results showed that the risk of hydro-political tension was expected to rise to 74.9% by 2050 and to 95% by 2100, particularly in the basins of the Nile River, Euphrates and Tigris Rivers, Ganges and Brahmaputra Rivers, and Colorado River. This study highlighted the importance of water cooperation efforts. Dresse et al., in 2019 [19], presented a theoretical framework to assess the inherent environmental challenges that act as a tool for cooperation and peace rather than violence and competition. The authors emphasized the lack of an environmental peacebuilding

framework. This study considered three phases (conditions, mechanisms, and outcomes) through three generic approaches (technical, restorative, and sustainable environmental peacebuilding). This framework, along with hydro-politics and socioeconomics, can enhance the understanding of the importance of environmental cooperation across borders to ensure sustainable peace. Drawing attention to the conceptual gaps where the systematic framework is limited by dynamic mechanisms and political processes, the authors argued that the framework requires multidisciplinary dialogue and an analytical matrix that assists both decision makers and scholars in considering the multidimensional components of any future framework dealing with environmental peacebuilding. Zeitoun et al., in 2020 [20], identified destructive forms of cooperation and strategic opportunities for transformation using an analytical approach. The study employed a hydro-social and hydro-diplomacy framework and suggested three ways to deal with hydro-political challenges: (1) resist analyzing the situation under the assumption that scholars who are not associated with the decision makers are likely missing the bigger picture, (2) provide the analytical team with the required individual networks and trust the analysis; and (3) risk transformative analysis to devise new approaches to managing the water conflict. Recently, Baranyai, in 2020 [21], provided a collective review of previous theories on water conflicts and cooperation. The authors divided the evolution of water conflict into three eras: early studies that focused on the conflict potential in transboundary basins, studies from the 1980s and 1990s that focused on the triggers for conflicts and water wars, and recent research on overcoming conflicts and enhancing cooperation that highlights the inherent complexity of water relations. A new proposal for water conflict theory was proposed that focused on the geographical and political variables that influence the behavior of riparian countries. This theory is based on geography and water availability, geopolitical setting, level of economic development, sovereignty, domestic issues, capacity shortages, and cultural factors.

*1.2. Water Nationalism and Water Transitional Competition*

Binding water with nationalism is an arising discourse that justifies the construction of dams, where some researchers have discussed the water nationalism notion. Menga, in 2015 [22], explored nation-building and the interventions between water and power. The author highlighted how decision makers could use the dam symbolism to gain legitimacy and boost a sense of national identity and patriotism. The gigantic Rogun Dam in Tajikistan is used as a case study to demonstrate the Tajik government's behavior in creating and strengthening a nationalistic discourse and promoting the dam as a patriotic project. Lately, Wheeler, in 2021 [23], described the notion of water nationalism as a "desire by states to gain maximum national advantage from the exploitation of their natural resources". The author presented the impact of water nationalism and water identity on the academe of water studies affected by misinformation by politicians, mass media, and textbooks. The research emphasized the growing water nationalism by nationalist parties claiming sovereignty and safeguarding water resources, which eventually influenced the water policies. The outcome shows that water nationalism extends far beyond governments to civil society and academic institutions driven by national identity and a sense of competition over common pool resources. The research used two case studies Jordan Basin, and the Nile basin concluded how water nationalism is entrenched and flagrant among academic researchers. Daoudy, in 2010 [24], illustrated the role of water and power in shaping regional dynamics and how the geographic position, economic and military privileged the upstream countries. The author demonstrates the geopolitics of water in the Middle East considering the Euphrates–Tigris basins. The research presents the past legacy, the inherited perceptions, and the relationship between Turkey (upstream country), Iraq, and Syria (downstream countries). The outcomes show how the emergence of Turkey as a regional power using a mega water project Southeastern Anatolia Project (GAP) as a strategic element to enforce the regional political agenda. Hussein, in 2019 [25], showed how the impact of water scarcity discourses on transboundary water governance has been overlooked. The author used three transboundary water governance cases: the Yarmouk

River, the Jordan River, and the Disi Aquifer. The outcome shows the gaps in explaining the discourse of transboundary water governance, especially for Arab regions where most of their surface waters originate outside their countries. The research explores the needs of the broader context and considers regional geopolitics and power asymmetries.

Between 1900 and 2017, over 330 water cross-state boundary disputes were recorded and mostly remained unresolved [26]. However, the water transitional competition may take different forms, and the four most familiar cases are as follows:

(a) Dam building or water impoundment by the upstream country: The Euphrates–Tigris basins in this context are a good example where Turkey as an upstream country impounded the water flow for the Two Rivers considerably by the huge dam complex through the GAP project, and the water flow reduction was substantial, particularly for Iraq, as the ultimate downstream country where total water flow for both Euphrates and Tigris Rivers will be reduced by approximately 50% by 2030 [27]. Another example is the water transboundary conflict in the Nile basin. The upstream country Ethiopia built the Grand Ethiopian Renaissance Dam (GERD) to generate hydro-power, where 86% of the Nile water is contributed by Ethiopia. The 74 billion/$m^3$ storage capacity dam is intended to generate 6000 MW for Ethiopia. However, the dam led to a major water conflict with the downstream countries, Egypt and Sudan, who considered the GERD project a threat to their national security. Several rounds of negotiations ended with unclear agreements between the riparian countries [28]. Recently, Hussein, Conker, and Grandi, in 2022 [29], conducted a comprehensive literature review to investigate how the state's elites justified large hydraulic projects such as massive dams. The authors determined four reasons to answer that question: extend the state's power/influence abroad and control the domestic level, promote a national pride symbol, elevate political relations, and develop the economy. The research also addressed the fact that massive hydraulic projects might be considered a handy tool to enforce a foreign policy by the state.

(b) Diversion of waters from a natural body of water: The dispute between Bolivia and Chile over the Lauca River. In 1962, Bolivia claimed that Chile, an upstream country, diverted the Lauca River to irrigate the small Sobraya and Azapa valleys for food production in the northern desert. The Lauca River diversion has led to political tension between Bolivia and Chile, and the dispute is still unresolved [30]. Several diversion projects on the Karun and Karkheh Rivers established by Iran for irrigation purposes led to the complete removal of over 40% of the Shat-Al-Arab supply in Iraq. The expected purpose is to force Iraq to negotiate a settlement for the Shat Alarab issue [31,32].

(c) Upstream pollution from industry or mining operations: The transboundary basin of the Kura-Araks River Basin is shared by Georgia and Turkey as upstream countries and Armenia, Azerbaijan, and Iran as downstream countries. Both rivers join in Azerbaijan before entering the Caspian Sea, where more than 31% of the Kura-Araks River Basin is located in Azerbaijan [33]. Pollution on a large scale across the basin is deducted, where a considerable volume of effluent is discharged, affecting the surface water and groundwater. The major source of pollution is municipal wastewater, which carries organic matter and toxic substances downstream countries [34]. The pollution led to political tension between the riparian countries [35]. The transboundary Lake Victoria Basin is shared by five African countries, i.e., Tanzania, Kenya, Uganda, Rwanda, and Burundi, though half of the basin and lake areas are located in Tanzania. The basin suffers from poor fishery and irrigation practices, and the water quality of Lake Victoria deteriorates, whereas concentrations of chemical wastes are increasing and evidence of mining activities are indicated [36]. This improper water utilization may cause potential conflicts among the users [37].

(d) Foreign control over water distribution: The resistance of Cochabamba people in Bolivia against water privatization is an example of a foreign international consortium, the Aguas del Tunari, which is mostly owned by the private sector of the USA, UK, and Spain and was granted as a sole bidder for water supply, sanitation services, and irrigation. The rate of the water supply has been drastically raised [38], which led to the water uprising day in April 2000 against water privatization to reclaim public control over water utilities [39].

In this context, McDonald and Swyngedouw, in 2019 [40], and March et al., in 2019 [41], proposed that water service remunicipalization will continue growing and that it should be handed back into public control due to public dissatisfaction with water privatization. On the other hand, the private sector and powerful multilateral actors will resist water service remunicipalization, along with support from international financial institutions, which would restrict the growth of the water services remunicipalization concept. The authors provided new case studies, such as the political forces and social movements in Barcelona, identified key stakeholders to support their hypothesis, and highlighted the way that the investigated actors influence local and international policymaking.

The current study focused on type-a water conflicts that are caused mainly by dam building/water impoundment by the upstream country. Highlighting the large dam influence, Khagram, in 2004 [42], traced the changes in non-governmental entities as transnational coalitions from passive to active actors that espouse environmental protection and mitigate the risk of big dam consequences, such as social and environmental implications. The author considered the huge dam complex of the Narmada River Valley Dam Projects in central India as a case study and compared it with other international dams worldwide to highlight the transnational struggles of large dams. Later, Khagram and Ali, in 2006 [43], focused on the linkages between environmental change and conflict and promoting peace. They claim that limited research has been conducted on the environmental effects of violent conflict and war on traditional and security institutions, such as militaries and military-industrial complexes. Rigorous research on the consequences of peace or human security for the environment is virtually non-existent. Scudder, in 2012 [44], provided a comprehensive assessment globally by examining the political costs, benefits, and risks of the development of large dam projects and questioning the suitability of the current development paradigm for large dams. In his next contribution, Scudder, in 2018 [45], conducted a long-term analysis of the negative impacts of large dams on different continents, the Middle East, and Africa within three sets of periods, i.e., 1956–1973, 1976–1997, and 1998–2018. The author explained that his perceptions of large dams evolved over time and how the perception turned from a unique opportunity to have integrated development for the river basin versus the actual cost to socio-economic and environmental factors. Han and Webber, in 2020 [46], offered a geopolitical interpretation of three dams built in Ghana during the Cold War. This study examined the hydro-political relationships between national and international actors, with China trying to expand its international soft power against British and American anti-communist efforts. This research illustrated the evolving technical, political, and economic intricacy of large dams.

*1.3. The Current Research and Objective*

The present research demonstrated case studies of inter-state water transboundary conflicts/cooperation, along with the possible actions, spiral escalation, and potential internal risks that threaten the actors at the state level. Despite past research, a broad, structured theory that covers all possible collective events involved in hydro-politics has not yet been established. Thus, clarifying the conceptual process and establishing systematic patterns would help the researcher better understand hydro-political events. The author of this study analyzed the basic concepts of this theory in 2019 [47]. However, the current research offers a sophisticated framework for establishing a structural theory of hydro-politics. The current new framework has not previously been presented, where results have been interpreted and discussing possible implications. The author is expected to add value to the accumulated knowledge in theorizing the water-politics area, which has been collectively presented in the literature review and the subsequent studies in the last two decades. This theory is designed to address some key issues associated with global water conflicts, such as mismatches in collaboration regarding hydro-political events, whether adopting a policy of international relations affects the actions of related parties, and whether negotiations over water resources represent negotiations over the future of the countries

and regions involved. In addition, the current theory can be used to identify the level of conflict, its expected consequences, and the possible effects of individual actions taken.

The objective of this research is to establish a structured theory of hydro-politics associated with hydro-political events caused by transboundary water control in an upstream country. This framework aims to produce a model that can be applied on a global scale. The proposed theory may help (1) model potential conflict scenarios and events to determine hydro-political consequences; (2) highlight the impact/contribution of international relations doctrine as an approach to fostering peace or escalating conflict; and (3) reveal the perception gaps within the different conflict stages between riparian countries that can help decision makers understand the risk of conflict and turn challenges into opportunities.

## 2. Materials and Methods

The framework for the present study was designed in the form of water impoundment and building dams in the upstream country. The framework uses a process that reflects the water conflict actions and consequences in the real world as following: (a) identifying at least two actors representing upstream and downstream countries to demonstrate possible scenarios that lead to gradual escalation and the repercussions that arise as a result of that escalation, with both actors driven by political aspirations to control and maintain the water resources; (b) In addition to global challenges that work as key pressure factors to aggravate the conflict chances within the complex system. Two major factors are discussed to disclose the influence of the water conflict between the actors or encourage them to cooperate: political perspectives that decide to develop the water projects on the watercourse and expected actor-action implications on the downstream countries that considered additional pressure components to trigger the water conflict, indicating the positive pressure from the global efforts to establish common grounds for cooperation; and (c) visualizing four components within the complex system through interrelated events for the actors (the upstream and the downstream countries): a description of the water situation between the actors and whether it reflects a conflict or cooperation status, the level of the water conflict and escalation, the type of power balance between the upstream and downstream countries, and the expected hydro-political consequences that would propose the form actor's reactions. These events formed the components of each phase, followed by a series of hydro-political scenarios. Figure 1 shows the initial components of the eight phases of the theory.

A comparative approach was used to investigate, analyze, and integrate information. The secondary data have been collected from modern literature, academic institutions, scientific articles, reports through the Google Scholar engine, government statistics, industry associations, private sector webpages, and public portals. In total, 216 international articles, documents, supportive documents, such as executive agreements, and reports were reviewed and evaluated in detail. These sources have been classified later for the following sectors: (a) theories of hydro-politics, (b) water conflict and resolution, (c) water and international relation theories, and (d) water governance and policies. About 35% of the studies were excluded because they were irrelevant to the current research or did not meet the above selection criteria. A total of 26 case studies covering international, regional, and local contexts have been used. Five case studies have been presented in the introduction, reflecting water pollution, privatization, and diversion. However, 21 case studies relevant to the water conflict based on water impoundment and building dams were systematically reviewed, interpreted, and integrated with the proposed theory to support the hypothesis in each of the eight phases. These global case studies cover the Far East, the Middle East, Europe, Africa, South America, and North America. A map illustrating the 21 case studies is generated by downloading the shapefile of the world from the public domain of World Administrative Boundaries—Countries and Territories [48], after which ArcGIS 10.8 has been used to layout the map. Primary data through Twelve group discussions were conducted, focusing on international water conflicts and their resolution (eight held online and four face-to-face) from August 2020 to September 2021. In

total, approximately 410 specialists from the water-related academic, executive, political, and/or non-governmental organizations participated.

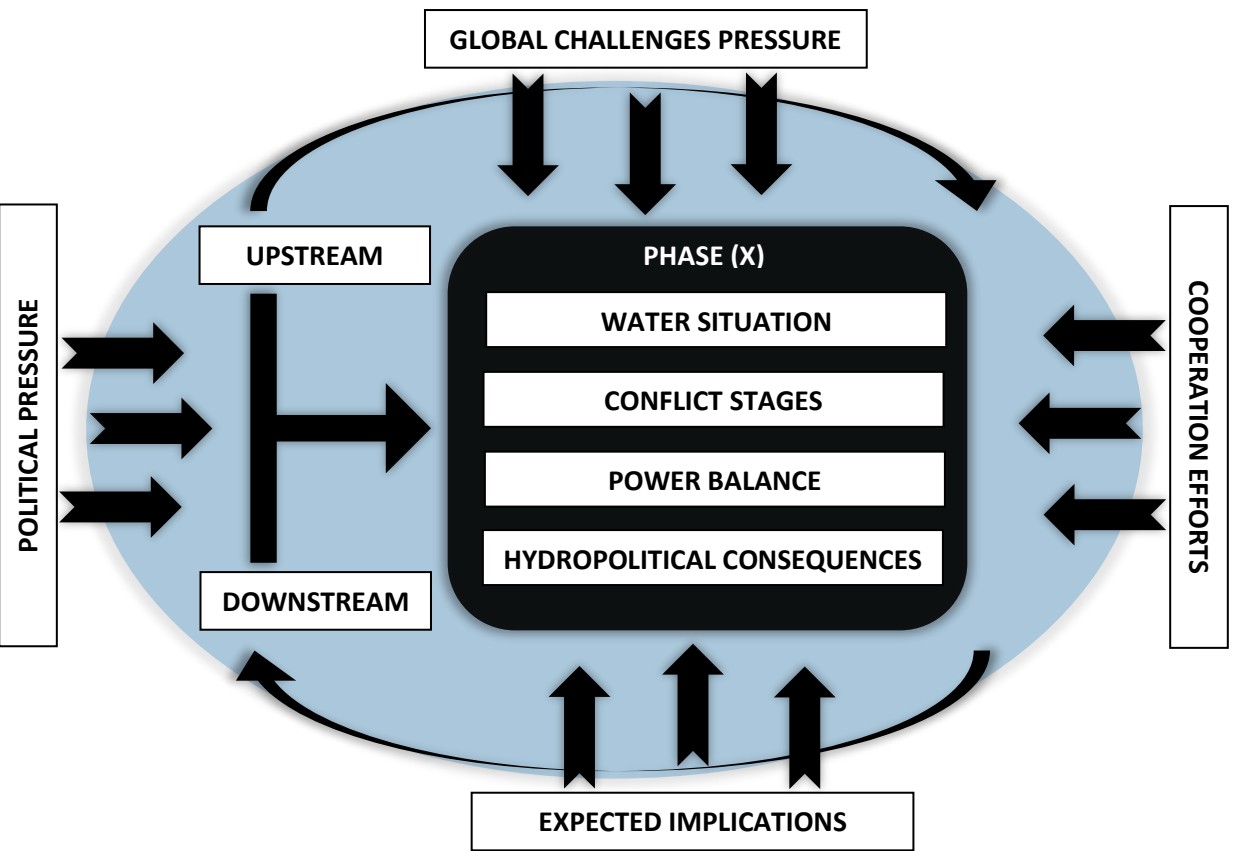

**Figure 1.** The components of the hydro-politics theory and the framework of spiral escalation of water conflict.

Two different approaches were integrated into the proposed theory to simulate water conflict escalation. The concept of conflict stages has been used in past research to describe the struggle, discord, and disagreements between actors as a series of phases, whereas the organizational conflict concept and models from Pondy [49] and Williams [50] have also been used. Investigating the connection between international relations theories and the water transboundary would help reveal the root cause of water conflicts. The wide range of possible frameworks would provide a social dimension to the complex system and explore the form and relationship between humans' possible actions and reactions, the capacity of natural resources (in particular, water), and geopolitical aspects. Six international relations theories (classical and critical theories) were incorporated to demonstrate the influence of the adopted political doctrine of practicing hegemony [51] by controlling water to promote state power abroad, explain the economic dimension as a key driving force by impounding water resources through Marxism [52], disclose the inevitable conflict over the water resources by understanding the key elements of classical realism [53,54], address large-scale dams as an anarchic source to escalate the water conflict supported by the structural realism [55], illustrate the role of epistemology and the intercultural barrier that expand the gap of communication between the riparian countries guided by the theory of constructivism [56], and discover the way forward to have a rational scope to manage the inter-state transboundary water by employing the global governance [57]. Each theory was used to explain a particular influence on a potential conflict event. The open-source program Plectica was used to build the final model for the proposed theory and illustrate the overall framework [58].

## 3. Results and Discussion

Water sustainability is considered a complex problem [59], and hydro-politics in particular is a transboundary challenge that includes various factors and interactions [60]. Globalization means that most inter-state conflicts or regional actions have a ripple effect internationally. Thus, it is vital to discuss and adapt to transitional challenges. There are several global water-related concerns, but the most serious is climate change, which is a cross-border challenge that can cause socioeconomic and environmental disruptions [61]. For example, approximately 35 million people have become climate migrants in Bangladesh, and by 2050, this number is expected to increase to approximately 150 million [62], whereas 4.8–5.7 billion people will live in water-scarce areas. In addition, 25% of the agricultural sector's cultivation damage is due to climate change-related disasters, particularly in developing countries [63], and the current global goal is to limit global warming to no more than 1.5 °C above pre-industrial levels [64]. In addition to the impact of climate change, the global population will reach 9.7 billion by 2070 and 10.9 billion by 2100 [65], which is a severe concern in terms of the global economy and the supply of natural resources. Achieving integrated water resource management (IWRM) by 2030 is a fundamental component of the UN's Sustainable Development Goals (Indicator 6.5.1); however, the current level is only approximately 50% globally, and doubling this is urgently needed [66].

Developing leadership in water conflict resolution is crucial to achieving sustainability. The presence of knowledgeable leaders creates the momentum required to better understand watercourse conflicts, leading to less litigation and more cooperation between riparian countries. Given the looming water shortage crisis, this leadership style is still lacking. Furthermore, many water-related issues represent a failure in governance and policy development [67], thus immediate actions in this respect are required to secure global water access for drinking and hygiene [68]. Investing in technology is also a crucial strategy to bridge the gap between water supply and demand and solve issues related to water quality. The agriculture sector consumes the majority of water from freshwater resources (~70%), and food production will need to increase by 60% by 2050 to cope with demand [69]; however, adopting modern irrigation technology can reduce water use by up to 50% [70]. In contrast, the level of public and state awareness of the tension and conflict over water supplies and their consequences still needs to be raised. Nevertheless, global water-related crises and new norms associated with the COVID-19 pandemic have exacerbated the demand for water for hygiene and sanitation purposes [71], which may increase water tension at both the local and regional levels.

The theory of hydro-politics (Figure 2) proposes primary conditions for any two actors (upstream versus downstream) driven by political decisions. The structured theory leads to a model with eight phases that can be applied to any case of watercourse/basin tension worldwide. Each phase describes (a) the form of the relationship between riparian countries in terms of transboundary water, (b) the stage of the conflict, (c) the power balance between riparian countries, and (d) the expected consequences of hydro-political actions. Multiple actors across a basin are common, potentially complicating the model. However, this approach simplifies the process and visualizes broad events that can lead to a set of possible actions and consequences.

### 3.1. Water Harmony (Phase 1)

This phase represents the ideal situation of excellent cooperation between actors, harmonious water relations, and no conflicts. This situation is coupled with a symmetrical power balance. Good communication between riparian countries and mutual interests leads to strategic agreements that collectively lead to an equilibrium system. Global examples of water cooperation are Portugal and Spain, which share five main river basins, where dozens of dams have been built on the Spain–Portugal basins [72]. Spain, as the upstream country, contributes ~70% of the basin water. In 1998, they were able to reach a strategic agreement by signing the Albufeira Convention aimed at harmony and equilibrium [73,74]. Another example is the SADC Revised Protocol on Shared Watercourses signed in 2000 by

Mozambique and Zimbabwe for the Sabi River Basin [75]. In 2014, heavy flooding caused a partial failure in the Tokwe Mukorsi dam that put at risk the lives of ~40,000 people [76,77]. This followed by the Pungwe Basin Bilateral Agreement in 2016 to adopt a prevention flood strategy and support institutional transboundary water management [78].

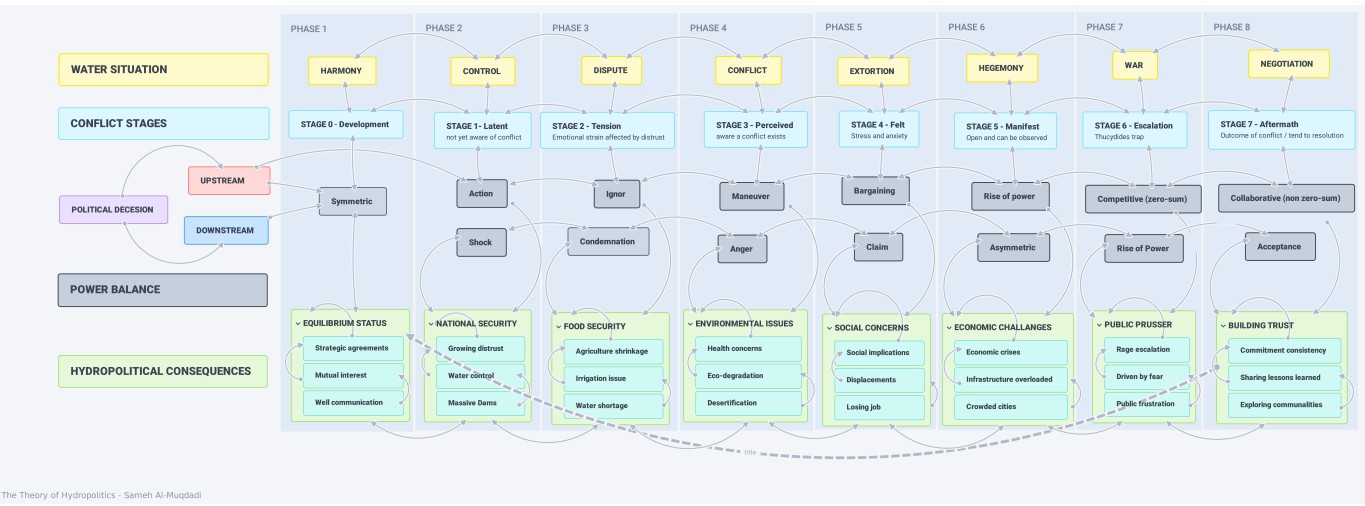

**Figure 2.** The theory of hydro-politics.

### 3.2. Water Control (Phase 2)

When an upstream actor decides to implement strategic water projects within a shared catchment (e.g., constructing a dam on a watercourse), the upstream actor usually justifies it as part of the national development plan, with development being the main driver. Under this justification, water resources are controlled, leading to a hydro-control relationship. The conflict is latent at this stage, and actors are unaware that conflict may exist. However, although this action may be justified as a form of national development by the upstream actor motivated by sovereignty, this may be a shock for the downstream actor because controlling water abuses the basic concept of justice, and this is the Harmon doctrine dilemma theory of water rights [79]. However, water control has progressed over time, starting with local community practice, shifting to the local government, then to foreign water privatization, and escalating to regional or international water conflict [80]. Regardless of the actual intentions of the upstream actor, the downstream actor is likely to believe that the building of dams has a negative intention to control natural water flow. Eventually, this will lead to growing distrust, which can create further complications and a growing sense of national and regional insecurity and increase the likelihood of anarchy in water relations. Hans Morgenthau, the founder of structural realism theory [55], introduced the idea of an anarchy system, in which there is no central authority to enforce policies and guide the relations between the actors, which aligns with Warner et al.'s [81] assumption that transboundary water relations are nonlinear. This anarchy comes with a cost represented by a security dilemma. In particular, the downstream actor will be insecure, be obsessive about survival, and doubt the chances of cooperation. The construction of massive dams would directly feed this anarchy system.

Although there are a few limitations in the United Nations Convention on Non-Navigational Uses of International Watercourses [82], the driving principle behind this convention is to encourage cooperation [83]. Article 5 emphasizes the principles of equitable and reasonable utilization and participation, whereas Article 7 highlights the obligation to not cause considerable harm. In addition, Article 12 dictates that the upstream actor should notify the downstream actor in a timely manner prior to the launch of any potential water project for an international watercourse and provide appropriate information and environmental assessment.

However, whether these requirements are fulfilled remains unclear, as seen in a number of previous examples. The Iliso Dam was constructed by Turkey in the Tigris

Basin and opened for operation in 2018. This is one of the most critical concerns for Iraq as a downstream country. Although the Iliso Dam will produce ~3800 GWh/year of clean hydro-power [84], the dam is expected to reduce the water supply to Iraq from the Tigris River by up to 25% [85]. In addition, the construction of the Rogun Dam in Tajikistan on the Amu Darya River had been stalled since 1976 because of the collapse of the Soviet Union. In 1998, a strategic partnership was signed by the Government of Tajikistan with the Governments of Kazakhstan, Uzbekistan, and Kyrgyzstan [86]. The construction plan was launched again in 2016 in the hope that the dam would help alleviate the country's energy shortages by generating 3600 MW per year. However, downstream Uzbekistan is concerned that this dam will reduce the water available for irrigation purposes, which has led to distrust and fear in the relationship between both countries. This was alleviated when the Uzbek president announced that the project would improve bilateral relations [87], highlighting the great power of political statements in this context. Another water control example is the Nile basin, where Egypt accuses Ethiopia of hostage the Nile River through the Grand Ethiopia Renaissance Dam which becomes a source of tension between the riparian countries of the Nile basin [88].

### 3.3. Water Disputes (Phase 3)

Affected by Phase 2, controlling water using massive dams leads to a shortage of water resources. In particular, it can severely disrupt the irrigation plans of downstream actors, harming the agricultural sector. Thus, for downstream actors, these dams are perceived as threats to national food security. This type of dispute between the actors is common at this stage, with tension rising due to the emotional strain arising from distrust. The downstream actor may condemn the actions of the upstream actor, while the latter ignores or denies these concerns. However, reaching an agreement remains possible because the dispute is a short-term disagreement, and healthy negotiations between actors are likely to end with some resolution.

The Indus Basin water dispute between India and Pakistan dates back to 1948, when India controlled the flows in West Punjab. This increased tension between the two countries, and it took six years of negotiation to sign the Indus Waters Treaty in September 1960 [89]. In addition, India and Bangladesh share 54 rivers. For example, the Farakka Barrage, established in 1975 on the Ganges River, has caused considerable hydrologic changes to the downstream basin and is subject to water disputes [90,91], which continued for 30 years, until the Ganges Water Treaty was signed in 1996 after a series of negotiations [92]. Iraq and Turkey have a long history of water disputes over the Euphrates–Tigris basins; the water disputes date back to the 1920s after the collapse of the Ottoman Empire. Iraq, Syria, and Turkey emerged as independent states sharing rivers, which created an environment for potential conflict [93]. Back to the Nile basin, the diplomatic efforts fail to resolve the dispute between Egypt and Ethiopia over Nile water sharing [94].

### 3.4. Water Conflicts (Phase 4)

The possibility of conflict always exists when actors have different requirements, interests, or perspectives. If negotiations lead to a dead end, the next level of the water relationship between the actors will be conflict, and both actors will be aware that this conflict exists. Unlike in Phase 3, this conflict is a long-term disagreement that is non-negotiable. The hydrogeological consequences at this stage are more critical, with downstream actors likely to suffer from desertification, ecological degradation, and health concerns. These environmental issues can lead the downstream actor to express their anger, whereas the upstream actor will continue to deny or ignore the consequences or even blame the downstream actor for their poor internal water resource management. Time plays a critical role here, and it will be advantageous for the upstream actor to stall the conflict until their hydro-projects are complete. In contrast, the downstream actor will continue to suffer environmental implications if no legal action is taken to suspend these projects.

After decades of rising tension, Egypt, Sudan, and Ethiopia reached a dead end in their negotiations over the Grand Ethiopian Renaissance dam in the Nile Basin. This tension shifted to long-term disagreement between Egypt and Ethiopia, and conflict emerged at different times, including aggressive political statements and potential military action [95]. However, one of the reasons that the Nile Basin did not devolve into a water war was the power balance, with Ethiopia (the upstream country) being more geographically powerful and Egypt having a greater military capacity [96]. Another example of water conflict is between Iran and Afghanistan (the upstream country) over the Helmand River Basin and the Harirud-Murghab River Basin. Although a water treaty over the Afghan-Iranian Helmand-River was created in 1973 [97], the unilateral act of building the Kamal Khan Dam is stimulating the water conflict between the two riparian countries [98]. The water control by Afghanistan led to considerable drought and desertification. As a result, over the last two decades, approximately 25–30% of the population has left the Sistan region to move to cities due to water shortages [99]. The depletion of Lake Chad in the 1960s, which was an essential icon of agricultural heritage, is another face of water conflict between four neighboring countries, i.e., Chad, Nigeria, Niger, and Cameroon, where hydroelectric dams have been built [100]. The Tiga and Challawa Gorge dams built by Nigeria led to water conflict with the downstream country Niger [101]. Lately, Nigeria's planned to join the UN Water Convention, a matter that could be a tipping point for cross-border water cooperation in Lake Chad and Niger basins [102]. The completion of the Southeastern Anatolia Project (GAP) on the Euphrates–Tigris basins will make Turkey withdraw ~70% of the Euphrates' water, which is likely to strain relations between Iraq, Turkey, and Syria and put the region in danger of inter-state conflict escalation [103].

*3.5. Water Extortion (Phase 5)*

If the interests of the two actors cannot be reconciled in Phase 4, the water scarcity for the downstream actor and the associated environmental issues that arise in Phase 4 can potentially lead to social concerns at the local level, escalating further to the regional level. This can include the loss of jobs, particularly for farmers and others involved in agricultural activities. This can cause large-scale migration to large cities and have other social implications. An example of the local level is the displacement of 2180 families because of the construction of the Bhakra Dam in India [104], whereas 10 million people were relocated because of the Three Gorges Dam in China [105]. The downstream actor will claim their rights at this stage, whereas the upstream actor may bargain to gain more time. At this level, the conflict stage is assumed to be driven by stress and anxiety, especially if the water relationship with the upstream actor is perceived as extortion (i.e., using water as a tool to negotiate other political or economic issues).

Competitive bargaining is acceptable to a certain extent during negotiations, but difficulty starts when competitive bargaining crosses the line and becomes perceived as extortion. Legally, there is a fine line between the two, and this distinction may rely primarily on the opponent's perceptions. Wendt, in 1995 [56], discussed constructivism in international relations and argued that states can act as irrational individuals with both culture and language shaping their perceptions. Coupled with miscommunication, the belief that extortion occurs can create a worst-case scenario for water relations between the actors. Because this is a complex and dynamic system, irrationality is highly expected within each phase of the theory of hydro-politics, except for Phases 1 and 8, which are conflict-free. In 1992, regarding the Euphrates–Tigris Basin, the former Turkish Prime Minister announced, "We do not say we share their oil resources, and they cannot say they share our water resources. This is a right of sovereignty. We have the right to do anything we like" [106]. This statement could be interpreted as an attempt to supply a drying region [107], but it could also be perceived as a form of extortion with the goal of trading water for oil, leading to objections by the downstream countries of Iraq and Syria. The Mahakali River Basin is another example of water extortion, where India impounded the water resources of the Mahakali River by dams, such as impounding

water for hydro-power generation by Sarada Barrage [108], and denied the Mahakali treaty signed in 1996 between India and Nepal. The downstream countries Nepal, Pakistan, and Bangladesh politically accused India of using the water resources of the Mahakali River as a blackmailing tool [109].

### 3.6. Water Hegemony (Phase 6)

Due to the internal pressure of the social repercussions of transboundary water disputes, the number of displaced people can increase either locally or regionally, with large cities becoming crowded and their infrastructure overloaded. This would put a critical burden on the national economy, leading to an economic crisis if no strategic agreement is reached between the two states, where displaced individuals may become wave immigrants if the issue is neglected. The power balance between the two actors tends to be asymmetric, and the upstream actor gains power through water control after obtaining sufficient time to finalize its water projects. This stage is open and observable. At the same time, the notion and characteristics of hegemony may emerge, as reflected in the water situation between riparian competitors. The completed hydro-projects would lead to new hydrological maps for the basin, which would lead to a new norm of bargaining power. Thus, the bargaining power of the upstream actor dramatically increases. At this tipping point, it is unlikely that the upstream country will have to abandon its water projects.

Based on the neoliberal institutionalism theory and transnational relations, Keohane and Keohane, in 2005 [51], described hegemony as political domination in terms of the capacity of resources and the international political system. However, this phase also highlights the importance of local/regional political and economic pressure, which may drive decision makers to make difficult choices. Indeed, Karl Marx, who introduced Marxism as a key classical theory, singled out the economy as a key driver of conflict [110].

China and Myanmar have used their strategic geographical positions within the Mekong River Basin to establish water hegemony by controlling the flow of water through dozens of dams, leading to conflict with downstream countries Cambodia, Laos, Thailand, and Vietnam. In 1995, both China and Myanmar refused to cooperate with the Mekong River Commission after downstream countries proposed that sustainability be pursued and that exploiting water resources be avoided [111]. Hydro-colonialism in the United States and Canada, which ended with the 1909 Boundary Waters Treaty [112], involving the indigenous population, is another example of a hegemonic relationship developed via the control of water resources using hydroelectric dams [113]. In this context, a gain in power through hegemony appears when regional power can unilaterally impose a water-sharing scheme on others [114]. In contrast, some studies have indicated that hydro-hegemony could be relatively diminished if the less powerful riparian state adopts a resistance strategy to counter hydro-hegemony, that is, the Yarmouk River (Syria–Jordan) and Nile River (Egypt–Ethiopia) [115]. Different levels of hegemony have been explored, where Turkish hydraulic ambitions and power strategy on the Euphrates–Tigris Basins are considered as domestic and international hegemony applied to the water sector [116].

### 3.7. Water Wars (Phase 7)

Thucydides, the general of war and father of the classic realism theory of international relations, believed that war was inevitable when an ambitious rising nation threatened other powerful nations, because war is driven by fear [53]. Allison, in 2017 [54], described this concept as the 'Thucydides Trap', where an emerging power threatens other regional powers as international hegemons. Thus, Phase 7 represents this trap, with the rising power of the upstream actor being confronted by escalation from the downstream actor due to public pressure, particularly from water stakeholders. At this critical point, a toxic cocktail is offered when actors are under severe pressure, and at least one adopts the realism doctrine. This competitive environment reflects a zero-sum approach, which would lead to an inevitable war. This does not necessarily have to be an armed military war; rather, it may be a proxy war or economic war.

The Algiers Agreement between Iraq and Iran was established to settle the conflict around the Shatt Al-Arab River, which has strategic importance for transportation and exports. The bilateral treaty signed in December 1975 [117] aimed for a political settlement, with Iraq wanting to end the Kurdish rebellion supported by Iran. However, a few years later, the Shatt al-Arab River became a key reason for the war between Iraq and Iran. The escalation started with sovereignty claims over the river and finally led to the longest conventional war in the 20th century, from 1980 to 1988 [118]. After 2003, Iraq was bogged down by political instability, causing the power balance to shift to Iran. In the last decade, Iran has caused water shortages in Iraq not only because of diverting rivers that supply the Shatt Al-Arab River without consultation [31], but also by building a dam complex currently in operation, where a strategic plan has been set to build 109 dams in western Iran along the Iraqi border within two years. These dams are designed for use in irrigation projects [119]. Most of these dams are small-scale, but they can disrupt the natural river flow in Iraq; thus, they collectively function as one major dam [120]. Consequently, there are concerns that another conflict can break out. In addition, the Turkish South-Eastern Anatolia Project, which aimed to build massive dams in the Euphrates–Tigris Basins, led to potential water wars between Iraq and Syria in 1975 [106]. Since the 1970s, Syria also used the issue-linkage strategy by supporting Kurdish rebels (the Kurdistan Workers Party) to place political pressure on Turkey to gain the requested water shares [47,121]. However, the Orontes Basin also witnessed how Syria used a bilateral interaction strategy to exclude riparian countries Turkey and Lebanon. Syria and Turkey built the Afrin dam and Reyhanlı Dam, respectively, while the proposed friendship dam between Turkey and Syria has been suspended due to the Syrian uprising in 2011 [122].

*3.8. Water Negotiations (Phase 8)*

Wolf, in 2002 [123], argued that one of the key lessons from history is that although water can cause disputes, it often promotes cooperation rather than conflict, even between bitter enemies. During the post-conflict period, water issues can be resolved by appropriate mediation. The actors, particularly the downstream actor, are encouraged to explore their commonalities, share the lessons learned, and guarantee consistency in any future commitment. This phase involves trust building, where the upstream actor believes in a collaborative approach (i.e., a non-zero-sum game). The downstream actor will accept water projects in the shared catchment under certain guarantees and conditions to secure sufficient water resources. The conflict at this stage is in its aftermath, with the actors reviewing the outcome and seeking resolution. A strategic agreement may be introduced that acts as a springboard for a return to Phase 1 (harmony and equilibrium). However, this transformation from Phase 8 to Phase 1 relies on the leadership ability of the riparian actors, the type of international relations doctrine adopted, and the standard of the international security system, with the behavior of a state also driven by transnational and global threats, not just by the behavior of other states [124]. Rosenau [57], a founder of critical global governance theory, described the collective management of common transnational challenges that cross political borders. Global governance is characterized by non-traditional problems that are not expected to end at the political border of a country, such as a climate change, terrorism, weapons of mass destruction, and pandemics. These challenges require global efforts, regional collaboration, and a particular set of policies. The water conflict is a global challenge. The Adana Agreement in 1998 [125] represented a turning point in Syria–Turkey relations and ended the conflict before it descended into war. This strategic agreement was secured after a decade of cooperation at different levels and the doubling of the water supply for Syria [126], owing to the resilient international relations adopted by both countries at that time. This is similar to the Treaty of Versailles of 1919, signed immediately after the First World War; Article 358 of the Treaty declared that France had an exclusive right to use the Rhine River to generate power, but it had to repay Germany 50% of the value of the energy produced [127], where the new leadership embraced pragmatic international relations to start a new era and overcome the lean time of war. In 2021, after

decades of dispute over the Illiso dam on the Tigris basin, Iraq and Turkey officials announced signing a water protocol for the Tigris River and establishment of a joint research center for water [128]; although the protocol briefly mentioned that Turkey agrees to release "fair share of Tigris water", it does not address a solid agreement of the expected amount of the water allocations. However, the protocol is still considered positive progress between both countries in terms of water cooperation.

*3.9. The Spiral of Escalating Water Conflic*

Table 1 and Figure 3 show the 21 case studies considered in the present research, along with the details for each case, location, expected hydro-political phase, and key events. The case study of the Euphrates–Tigris Basins passes through all the proposed phases except phase 1, where water harmony is assumed hard to achieve due to the geopolitical dynamics and regional instability. However, the Nile basin also reflects at least three phases: water control, water dispute, and water conflict. The negotiations between riparian countries of the Nile Basin still face challenges to achieving a strategic agreement where each riparian country, particularly Egypt and Ethiopia, insists on not changing their positions. Where Egypt claims the historic rights over the Nile River and asks to fill the dam gradually within several years to avoid the reduction of water flow, conversely, Ethiopia claims no harm to the downstream countries will occur if the dam fills fast.

**Table 1.** The 21 research case studies and expected phases.

| # | Continent | Upstream | Downstream | Transboundary Water | Dam | Expected Phase | Key Event |
|---|---|---|---|---|---|---|---|
| 1 | Europe | Spain | Portugal | Minho-Sil, Douro, Tagus and Guadiana river basins | Dozens of dams | 1 | Albufeira Convention 1998 [73]. |
| 2 | Africa | Zimbabwe | Mozambique | Sabi River Basin/Pungwe Basin | Tokwe Mukorsi Dam | 1 | SADC Revised Protocol 2000/Pungwe Basin Bilateral Agreement 2016 [75] |
| 3 | Asia | Turkey | Iraq | Tigris River Basin | Iliso Dam | 2 | 2021 Tigris River Protocol and to establish a joint research center for water [128]. |
| 4 | Asia | Tajikistan | Uzbekistan | Amu Darya River | Rogun Dam | 2 | 1998 Fostering strategic partnership, signed by the Government of Tajikistan with the Governments of Kazakhstan, Uzbekistan, Kyrgyzstan [86]. |
| 5 | Asia | India | Pakistan | Indus Basin | Kishanganga Dam | 3 | Indus Waters Treaty in September 1960 [89]. |
| 6 | Asia | India | Bangladesh | Ganga river | Farakka Barrage | 3 | Ganges Water Treaty signed in 1996 [92]. |
| 7 | Africa | Ethiopia | Egypt and Sudan | Nile Basin | Grand Ethiopian Renaissance dam | 2,3,4 | Ongoing negotiation. |
| 8 | Asia | Afghanistan | Iran | Helmand River Basin and the Harirud-Murghab River Basin | Kamal Khan dam | 4 | 1973 the Afghan-Iranian Helmand-River Water Treaty [97]. |
| 9 | Africa | Nigeria | Niger | Lake Chad | Tiga and Challawa Gorge dams | 4 | Nigeria's planned accession to UN Water Convention could be a tipping point for cross-border water cooperation [102]. |

**Table 1.** *Cont.*

| # | Continent | Upstream | Downstream | Transboundary Water | Dam | Expected Phase | Key Event |
|---|---|---|---|---|---|---|---|
| 10 | Asia | India | Nepal, Pakistan, and Bangladesh | The Mahakali River Basin | Sarada Barrage | 5 | the Mahakali treaty signed in 1996 [109]. |
| 11 | Asia | China (town of Sandouping) | China (central China) | Yangtze River | Three Gorges Dam | 5 | 10 million people were relocated [105]. |
| 12 | Asia | Turkey | Iraq and Syria | Euphrates–Tigris Basin | Southeastern Anatolia Project (GAP) 22 dams | 2,3,4,5,6,7,8 | Ongoing negotiation. |
| 13 | Asia | India (Bhakra village) | India (district of Bilaspur) | Sutlej River | Bhakra Dam | 5 | 2180 families have been displaced [104]. |
| 14 | Asia | China and Myanmar | Cambodia, Laos, Thailand, and Vietnam | Mekong River Basin | 11 dams | 6 | 1995 Mekong Agreement [111]. |
| 15 | North America | USA (Minnesota) | Canada (Ontario) and the indigenous population | Rainy Lake | hydroelectric dams | 6 | 1909 Boundary Waters Treaty [112]. |
| 16 | Asia | Syria | Jordan | Yarmouk River | Wahdah Dam (Unity Dam) | 6 | 2003 signed a bilateral agreement [115]. |
| 17 | Asia | Iran | Iraq | Shatt Al-Arab River | 109 dam complex (Iran) | 7 | 1975 Algiers Agreement [117]. |
| 18 | Asia | Syria | Iraq | Euphrates Basin | Tabqa dam (Syria) | 7 | 1975 Iraq and Syria on the brink of war [106]. |
| 19 | Asia | Syria | Turkey/Lebanon | Orontes Basin | Reyhanlı Dam (Turkey), Afrin Dam (Syria) | 7 | 2011 Friendship Dam (Turkey—Syria) is suspended due to the Syrian uprising [122]. |
| 20 | Asia | Turkey | Syria | Euphrates basin | Atatürk Dam out of 5 total dams (Turkey) and Tabqa Dam (Syria) | 8 | Adana Agreement in 1998 [125]. |
| 21 | Europe | Germany | France | Rhine River | The Kembs dam supplies the Grand Canal d'Alsace | 8 | Treaty of Versailles of 1919 [127]. |

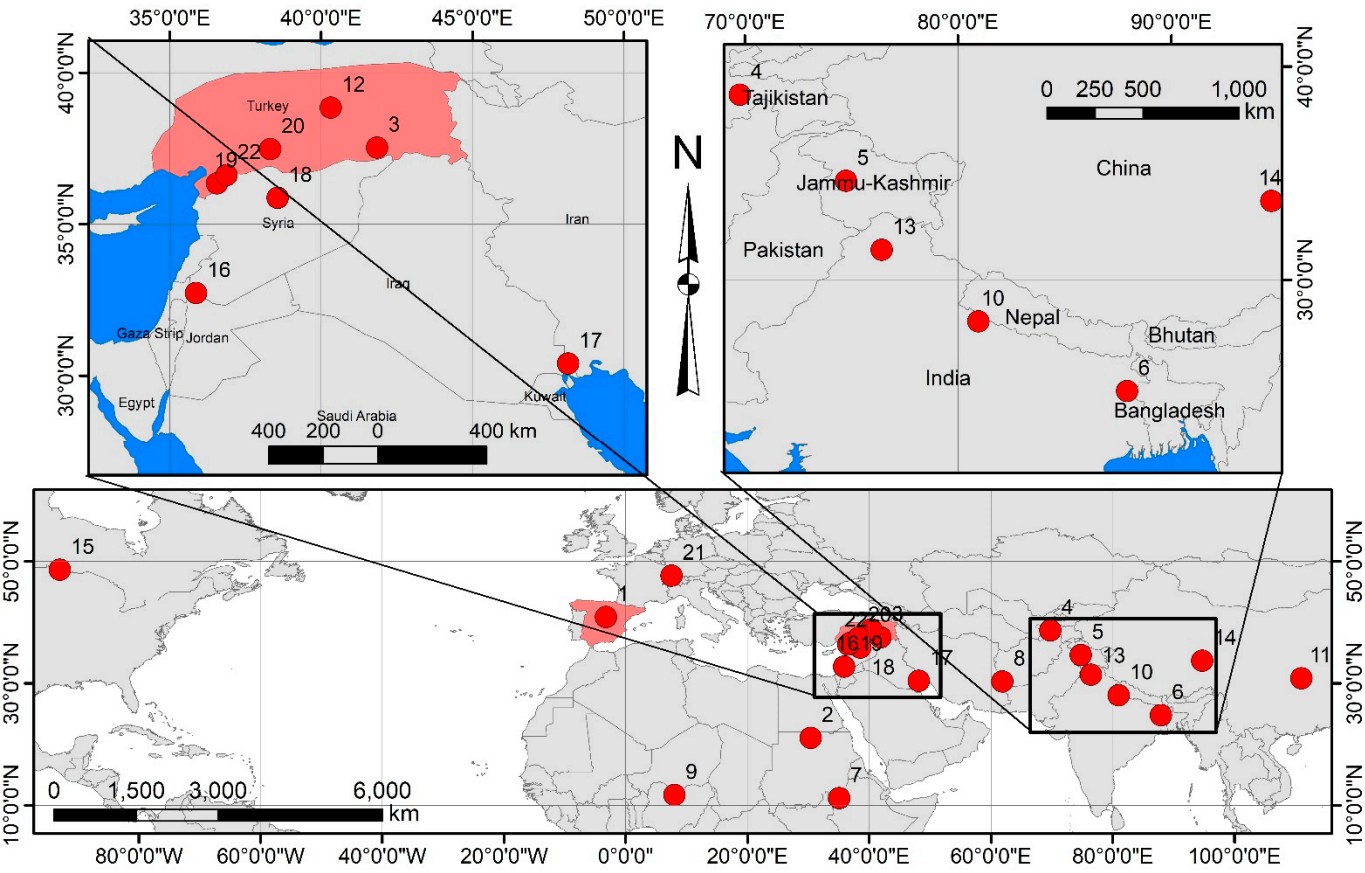

**Figure 3.** The 21 research case studies on the world map.

## 4. Conclusions

In the present study, eight phases were employed to produce a theory of hydro-politics based on water impoundment and building dams by the upstream country. These phases reflect the water situation between the riparian countries in different aspects: water harmony, water control, water dispute, water conflict water extortion, water hegemony, water war, and water negotiation, each phase representing a specific scenario that describes the relations between riparian countries over the transboundary water, the level of conflict, the power balance, and the expected consequences. These events represent a complex escalating system of water conflicts that runs from phase to phase in a ripple effect. This hydro-political spiral may move gradually or be suspended during any phase. However, it is clear that political decisions are highly influential across the entire cycle, with the potential to avoid implications or worsen the scenario, a matter that highlights the value of leadership and vision. Nonetheless, it is difficult to speculate about the scale of time on which this hydro-political spiral operates, because it depends on numerous variables in addition to the capacity of the key actors, the limitation of water resources, and the international relations approach adopted. However, it indicates that faster strategic agreements are reached, the better and more sustainable relationship between the actors over the water transboundary, in addition to the less serious expected negative implications. The results show a strong connection between the conflict/cooperation over watercourses and international relations theories adopted as a doctrine by the upstream and downstream countries, where six classical and critical theories have been integrated within the spiral phases: hegemony, Marxism, classical realism, structural realism, constructivism, and the global governance of institutionalism. An in-depth understanding of the interlinks between international relations theories and the behavior of riparian countries that dispute over the water transboundary is disclosing how water conflict challenges are politically oriented

rather than being technical ones. Understanding international relations theories beyond water conflict and cooperation would also increase the resilience of decision makers in terms of water conflict and transformation management and develop better environments to achieve agreement between riparian countries.

The contribution of the theory of hydro-politics and recognizing its components and events is as follows. It (a) helps uncover the dynamics and the root cause of water conflicts formatted in a structured framework, (b) reveals the gaps within the potential water conflict between riparian countries, (c) offers a particular scale that demonstrates the size of the challenge that actors face, and (d) assists decision makers in analyzing the collective risk to alleviate the possible consequences.

Controlling water through dams can lead to a series of implications, grouped into different phases, if no political decisions are made. Damage to food security, the environment, and socioeconomic levels can severely harm downstream countries and lead to internal public pressure that promotes state/political pressure and a higher risk of escalating conflict with upstream countries. This issue poses a regional and global threat to peace. Some of the reasons why the principles of the United Nations Convention of International Watercourses are not upheld could be the lack of communication/vision, lack of leadership, differences in international relations approaches, and deep distrust between the riparian countries.

**Funding:** This research received no external funding.

**Data Availability Statement:** Not applicable.

**Acknowledgments:** The author is very thankful for the academic support from different reviewers and the academic institutions that hosted several rounds of discussions worldwide. To Ann Koontz for her kind encouragement during the research timeframe and Khamis Al-Jubouri for the productive discussion about the Water Hegemony notion as a specialist in International Relations. Special thanks to Sanad Organization for Economic Development for the logistics and organizing most of the group discussions. To Arsalan Ahmed Othman for his kind scientific contribution by generating Figure 3. The dedication of this work goes to the Iraqi Public Leadership program (IPLP) that has been hosted by the American University of Sharjah—UAE, and sponsored by Crescent Petroleum. Thanks to all the 10 cohorts, particularly cohort 7, for the inspiration, fruitful discussion and the dream for better Iraq. Special thanks go to Yass Alkafaji and Majid Jafar for the motivation and generous sponsorship of the IPLP academic program.

**Conflicts of Interest:** The authors declare no conflict of interest.

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
