# Peer review of "The Spiral of Escalating Water Conflict: The Theory of Hydro-Politics"

_water, doi:10.3390/w14213466_

Round 1

Reviewer 1 Report (New Reviewer)

The paper discusses hydropolitics as framework to understand transboundary water conflicts. It does a good review of the literature on hydropolitics and critical hydropolitics. I would suggest improving the paper by:

- including a discussion on water nationalism, as discussed by Jeremy Allouche, Filippo Menga, and Kevin Wheeler. how does nationalism justify the construction on dams? Kevin Wheeler uses for instance the case od teh GERD to discuss this (Wheeler, K. G., & al. (2021). Water research and nationalism in the post-truth era. Water International46(7-8), 1216-1223.). See also: Menga, F. (2015). Building a nation through a dam: The case of Rogun in Tajikistan. Nationalities Papers43(3), 479-494.

-The role of discourses in justifying and supporting construction of mega project is also important in hydropolitics; see the work of Marwa Daoudy on this as well as Hussam Hussein.

- Naho Mirumachi and Menga took the case of Tajikistan to analyse nationalism, discourses, and forces driving the construction of dams; this should be better discussed in the framework of the TWINS; see: Mirumachi, N., & Allan, J. A. (2007, November). Revisiting transboundary water governance: Power, conflict cooperation and the political economy. In Proceedings from CAIWA international conference on adaptive and integrated water management: Coping with scarcity. Basel, Switzerland (Vol. 1215).; Mirumachi, N. (2007, May). Introducing transboundary waters interaction nexus (TWINS): Model of interaction dynamics in transboundary waters. In Third International Workshop on Hydro-Hegemony (pp. 12-13); Zeitoun, M., & Mirumachi, N. (2008). Transboundary water interaction I: Reconsidering conflict and cooperation. International Environmental Agreements: Politics, Law and Economics8(4), 297-316. 

- How original is your paper? What is its originality? How does it sit in the relevant literature? This should be better spelled out in the introduction

Author Response

Dear Reviewer,

Thanks for your valuable comments and time considered reviewing the work, please find below the answers, where all the revised sections are written in red colour in the revised manuscript enclosed:

  • Including a discussion on water nationalism, as discussed by Jeremy Allouche, Filippo Menga, and Kevin Wheeler. how does nationalism justify the construction on dams? Kevin Wheeler uses for instance the case od teh GERD to discuss this (Wheeler, K. G., & al. (2021). Water research and nationalism in the post-truth era. Water International, 46(7-8), 1216-1223.). See also: Menga, F. (2015). Building a nation through a dam: The case of Rogun in Tajikistan. Nationalities Papers, 43(3), 479-494.

Answer: Thank you very much for your kind suggestions, the valuable references have been added to the introduction section lines 183-200.

  • The role of discourses in justifying and supporting construction of mega project is also important in hydropolitics; see the work of Marwa Daoudy on this as well as Hussam Hussein.

Answer: Thank you very much for your kind suggestions, the valuable references have been added to the introduction section lines 200-214.

  • Naho Mirumachi and Menga took the case of Tajikistan to analyse nationalism, discourses, and forces driving the construction of dams; this should be better discussed in the framework of the TWINS; see: Mirumachi, N., & Allan, J. A. (2007, November). Revisiting transboundary water governance: Power, conflict cooperation and the political economy. In Proceedings from CAIWA international conference on adaptive and integrated water management: Coping with scarcity. Basel, Switzerland (Vol. 1215).; Mirumachi, N. (2007, May). Introducing transboundary waters interaction nexus (TWINS): Model of interaction dynamics in transboundary waters. In Third International Workshop on Hydro-Hegemony (pp. 12-13); Zeitoun, M., & Mirumachi, N. (2008). Transboundary water interaction I: Reconsidering conflict and cooperation. International Environmental Agreements: Politics, Law and Economics, 8(4), 297-316.

Answer: Thank you very much for your kind suggestions, the valuable references have been added to the introduction section lines 82-90.

  • How original is your paper? What is its originality? How does it sit in the relevant literature? This should be better spelled out in the introduction

Answer: Thank you very much for your valuable comment, a paragraph about work originality has been added to the introduction section lines 312-316.

Reviewer 2 Report (New Reviewer)

This paper establishes eight phases of conflict and cooperation in the shared waters of upstream and downstream countries. He believes that the architecture can assist decision makers in analyzing risks and mitigating the negative impacts of water conflicts. A few comments for this article are provided below:

1. The preface of the first chapter cites a large number of articles, and explains the origin of water politics research. It is recommended to explain in sections.

2. It is phase in the text that there are 23 cases supporting the hypothesis of the 8 phases of this paper. Can it be expressed in a table? The proposed table includes information on upstream and downstream countries, names of rivers in shared waters, whether dams are being built, and in which phase of the 8 phases proposed in this article.

3. Can several cases in several of these phase be illustrated by showing the map?

4. Among the 23 cases, if any of the cases have gone through the 8 phases (or 5-7 phases) proposed in this article from start to finish? If there is, it is suggested to add another section to explain. I think it can increase the connectivity of the eight phases of the hydro-politics proposed in this paper.

Author Response

Dear Reviewer:

Thanks for your valuable comments and time considered reviewing the work, please find below the answers, where all the revised sections are written in red colour in the revised manuscript enclosed:

  1. The preface of the first chapter cites a large number of articles, and explains the origin of water politics research. It is recommended to explain in sections.

Answer: Thank you very much for your kind suggestions, the introduction has been explained in three sections based on your kind suggestion.

  1. It is phase in the text that there are 23 cases supporting the hypothesis of the 8 phases of this paper. Can it be expressed in a table? The proposed table includes information on upstream and downstream countries, names of rivers in shared waters, whether dams are being built, and in which phase of the 8 phases proposed in this article.

Answer: Thank you very much for the excellent suggestion, a table including the case studies have been added including the upstream and downstream countries, names of rivers in shared waters, dams, expected phase/s, and the key event with reference for each. Please find the table in line 751.

  1. Can several cases in several of these phase be illustrated by showing the map?

 Answer: Thank you very much for the excellent suggestion, a world map has been generated to illustrate the case studies. Please find the map in line 752.

  1. Among the 23 cases, if any of the cases have gone through the 8 phases (or 5-7 phases) proposed in this article from start to finish? If there is, it is suggested to add another section to explain. I think it can increase the connectivity of the eight phases of the hydro-politics proposed in this paper.

Answer: Thank you very much for the valuable suggestion, additional section has been added showing the cases that went through a series of phases. Please find that section in lines 737-750.

Round 2

Reviewer 1 Report (New Reviewer)

The references need to be written in line with the journals system, which is surname and then first name capitalised, not the opposite.

You also included the paper of Kevin wheeler in the text, but it does not appear in the bibliography

for this reason the authors need to double check all in text refs and make sure they all appear. In the bibliography

Author Response

Thanks for your time and the valuable comments, please find below the answers, where all the revised sections are written in red color in the revised manuscript round 2:

  • The references need to be written in line with the journals system, which is surname and then first name capitalised, not the opposite.

Answer: Thank you very much, the references’ names have been adjusted accordingly following your comment; kindly find them in Lines# 190,200, and 207.

  • You also included the paper of Kevin wheeler in the text, but it does not appear in the bibliography.

Answer: Thank you very much for the heads up, the reference has been added. Kindly find the reference added in line# 190 reference number 22. You will find it also in the references section line# 861.

  • For this reason the authors need to double check all in text refs and make sure they all appear. In the bibliography

Answer: Thank you very much, the references have been reviewed and adjusted based on the Water journal policies and instructions.

Reviewer 2 Report (New Reviewer)

The author is very careful to follow the previous suggestions to make changes. But there are still 2 questions and suggestions.

1. Please confirm how many cases are discussed in total? According to Table 1 and Figure 3 in the article, there are 21 cases, but the abstract says 23 cases.

2. The author of the first chapter is divided into 3 sections, but the text in each section is still not paragraph.

Author Response

Thanks for your time and the valuable comments, please find below the answers, where all the revised sections are written in red color in the revised manuscript round 2: 

  1. Please confirm how many cases are discussed in total? According to Table 1 and Figure 3 in the article, there are 21 cases, but the abstract says 23 cases.

Answer: Thank you very much for the heads up, this is to confirm the case studies that have been implemented to justify the hypothesis are 21 and this was a typo. Kindly find the correction made in abstract line# 18. However, considering the 5 case studies that have been presented in the introduction to reflect water pollution, privatization, and diversion. The total will be 26 but only 21 case studies for water conflict - impoundment and building dams have been used and integrated with the proposed theory to support the hypothesis in each of the eight phases. Kindly find this detailed under the methodology section paragraphs lines# 362-367.

  1. The author of the first chapter is divided into 3 sections, but the text in each section is still not paragraph.

Answer: Thank you very much for the comment. At the first round review, you kindly suggested having the introduction as sections to avoid having a long introduction section, at the same 1st round review the other reviewer asked to add 5 more valuable references. The matter that makes your suggestion is essential to follow. Hence, I have reviewed the entire introduction section, this gave me the impression of the three sections will make the audience more engaged gradually with the topic step by step, your suggestion along with the 2nd reviewer’s comments combined together to end up with the following three sections to have a smooth review: Theorizing the Hydropolitics, Water nationalism and water transitional competition, and the current research and objective:

1.1. Theorizing the Hydropolitics (Line#35): under this sub-section, I have demonstrated the efforts of two decades by literature and researchers who provided substantial efforts in this specific field. One more reference was added based on the 2nd reviewer’s suggestion to that section you will find it in paragraph lines # 82-90.

1.2. Water nationalism and water transitional competition (Line#182):: since the 2nd reviewers also suggested adding/discussing the water nationalism notion, I have added paragraph lines #182-214 which then smoothly fit with the other part coming after about the major reasons for water conflict were water impoundment through dams is one of them.

1.3. The current research and objective (Line#303): under this section, I have focused more on the present research, the youth concept, how it has been evolving, and the objectives. A paragraph to discuss the originality of the work has been explained in paragraph lines# 312-316.

This manuscript is a resubmission of an earlier submission. The following is a list of the peer review reports and author responses from that submission.

Round 1

Reviewer 1 Report

There is an interesting germ of an idea in this submission.  I like the idea of the “hydrological loop” although it is less a loop than a spiral of escalating conflict, and that movement or stasis along the spiral results from political decisions.   That said, this is not a publishable manuscript.  You have a great deal of work ahead of you to get to the point where your argument is sufficiently developed to merit publication.  

 One task that you will need to attend to before rewriting is to think more carefully about how the ideas of different international relations scholars actually contribute to the argument that you are attempting to build.  In this submission, you are seem to be name dropping without convincing your reader that the works that you cite are relevant to the model that you are trying to build.  You also neglect a number of scholars who have done important work on hydropolitics.  At a minimum, you should become familiar with the literature on the international water governance and on the politics of dam building.  See in particular, the work of Eric Swyngedouw, Thayer Scudder, Sanjeev Khagram, and Ken Conca.

Second, you need to clearly identify the forms that transnational competition for water can take and give us examples of each.  As I see it, international water can take the following forms:

Dam building or impoundment in the upstream country, as was the case with Turkey and Iraq, which reduces and changes the timing water flow to the detriment of downstream neighbors.

Diversion of waters from a natural body of water.  A good case is the conflict over transport of the waters of the Lauca River in Chile upstream from irrigating communities in Bolivia into diversion structures that carry it to the Azapa valley in Chile for irrigation and urban consumption.  

Upstream pollution from industry or mining operations that renders a water supply unfit for irrigation or human consumption in downstream countries.

Foreign control over water distribution within a single state, as in the case of water privatization in Cochabamba Bolivia.  

There may be others as well, but if you are going to limit your argument to the case of upstream dams, you need to be explicit about this and tell the reader the reasons for your decisions to do so.

Your selection of cases is somewhat haphazard.  First, you need to restrict your argument to transnational competition for water resources.   In this paper you mention some cases of internal competition and don’t clearly differentiate between international and regional competition for water.   So, a third task will be to assemble and organize a series of cases of water competition that cross national boundaries and to describe them in sufficient detail so that your readers understand WHY they contribute to your argument.  

In your conclusion, you state: 

“The results show the strong connection between the conflict/cooperation over the watercourses and the adopted international relations theories that have been adopted as a doctrine by the upstream and downstream countries, where six classical and critical theories have been integrated within the pillars: hegemony, Marxism, classical realism, structural realism, constructivism, and the global governance of institutionalism.”  

I am afraid that this conclusion is very weakly supported.  You have not shown that those engaged in water negotiations are subscribing to particular political theories, nor have you shown convincingly that the theories identified in this sentence satisfactorily explain the cases that you have discussed. You have not even clearly stated what you mean by hegemony.

Once you have done this preliminary research, data collection and organization systematically, you can begin to think about questions of presentation.  You will need to begin your paper with an introduction that lays out the central question you intend to address, why it is important, and how you intend to answer the question.  Then offer your refined literature review and a section on methods letting us know how you selected your sample of cases of water competition.     Then you can discuss the model sketched out in a revised and streamlined figure 3.  Figure 1 is not all that helpful, and figure 2 is absolutely unnecessary.  Figure 3 is the one that matters, but it needs work.  I would eliminate the row of “global challenges.”  It confuses rather than supports your argument.   Also, the use of the term pillars isn’t very helpful.  It would be better to think in terms of stages of conflict development or escalation that may follow from one another, even though as you argue, they do not inevitably progress from negotiation to open conflict.  

To conclude,  to make your argument work, you need to sit back and spend some time to clarify your argument and to support it well before thinking about publication.  

Reviewer 2 Report

The idea for this paper is certainly an interesting one and I was looking forward to reading it.   On reading though I found the paper incredibly difficult to follow and it did not engage me.  It seems to be a literature review (with some workshops) upon which the author has come up with some categories.   How do these categories help in predicting water conflict?  Who are they actually for?  

The author seems to take certain things at face value which must be questioned in a theory of hydropolitics e.g the assertion around SDG 6.5.1 and that IWRM levels need doubling.    IWRM is fundamentally weak when it comes to politics......how is this paper related to that?  Surely this paper actually shows how limited and weak IWRM is in this regard.